# Exploring Shopper’s Browsing Behavior and Attention Level with an EEG Biosensor Cap

**DOI:** 10.3390/brainsci9110301

**Published:** 2019-10-31

**Authors:** Dong-Her Shih, Kuan-Chu Lu, Po-Yuan Shih

**Affiliations:** 1Department of Information Management, National Yunlin University of Science and Technology, Douliou, Yunlin 64002, Taiwan; qq1214x@yahoo.com.tw; 2Department of Finance, National Yunlin University of Science and Technology, Douliou, Yunlin 64002, Taiwan; D10424003@yuntech.edu.tw

**Keywords:** consumer behavior, electroencephalogram (EEG) biosensor, attention and meditation, brain computer interface

## Abstract

The online shopping market is developing rapidly, meaning that it is important for retailers and manufacturers to understand how consumers behave online compared to when in brick-and-mortar stores. Retailers want consumers to spend time shopping, browsing, and searching for products in the hope a purchase is made. On the other hand, consumers may want to restrict their duration of stay on websites due to perceived risk of loss of time or convenience. This phenomenon underlies the need to reduce the duration of consumer stay (namely, time pressure) on websites. In this paper, the browsing behavior and attention span of shoppers engaging in online shopping under time pressure were investigated. The attention and meditation level are measured by an electroencephalogram (EEG) biosensor cap. The results indicated that when under time pressure shoppers engaging in online shopping are less attentive. Thus, marketers may need to find strategies to increase a shopper’s attention. Shoppers unfamiliar with product catalogs on shopping websites are less attentive, therefore marketers should adopt an interesting style for product catalogs to hold a shopper’s attention. We discuss our findings and outline their business implications.

## 1. Introduction

E-commerce, which has grown exponentially because of Internet technology, has induced changes at the market, industry, and economic levels, and has profoundly altered life, politics, and society [1]. Online network platforms are used globally to undertake various online services, and such platforms provide a robust means for generating income and, for an immeasurable number of consumers from around the world, to shop online [2]. Shopping websites display numerous products organized by category; despite being produced by different firms, and many of these products share similar features. However, these products, along with their information, are too great in number to be processed by the human brain because of a human’s limited cognitive capacity, thus causing the consumer confusion and low satisfaction [3,4,5]. In addition, shopping is a series of decision-making processes aimed at satisfying consumer needs. The focus of academic attention should be shifted to online shopping behavior because of the mismatch between excessive stimuli and limited brain capacity [6]. According to Drucker [7], “the objective of a business is to create and retain customers”. Information about customers largely concerns their consumption behavior, which involves the processing and selection of product information. 

Generally, retailers want consumers to spend more time shopping, browsing, and searching for products in the hope that they make a purchase. On the other hand, consumers may want to restrict their duration of stay on websites due to a perceived risk of loss of time or convenience [8]. Previous research discussing the purchasing decision process has assumed that shoppers face time pressure. For example, Lin and Wu [9] found that time pressure will increase the proportion of consumers unable to make a judgment or a choice. Vermeir and Kenhove [10] suggested that consumers under high time pressure search less for coupons and products with a promotion. Rieskamp and Hoffrage [11] demonstrate that compared to those under low time pressure, individuals under high time pressure accelerate the search for information, using less information, and staying focused on the most important features. Liu et al. [12] indicate that when shopping online under time pressure, participants’ observation length and count for browsing products with high brand awareness were respectively longer and higher than those for browsing products with low brand awareness. However, when they shopped online without time pressure, no difference between products with high and low brand awareness levels was observed.

Time pressure is an essential variable of consumer behavior; it prompts a decision to be made within a limited time [13]. Moon and Lee [14] perceived time-pressure purchases as a consumption decision made within time constraints specified by the consumer, suggesting that time pressure indicates a sense of psychological urgency. Moreover, consumers typically base their purchasing decisions more on limited knowledge than on careful deliberation and comparison; such decisions tend to be made in a matter of seconds [15,16,17]. Pieters and Warlop [18] showed that time pressure affects visual attention in ways such that consumers skip certain brand elements to optimize their decision-making. Moreover, both the cue utilization model proposed by Olson and Jacoby [19] and the theory of planned behavior proposed by Ajzen [20] assume that consumers are aware of their purchase motives and can distinguish products and brands they intend to purchase. Thus, some consumers have selection criteria that form a basis for evaluating product brands and selecting the top-ranked ones.

Shoppers behavior online has been reported in many studies, and the browsing behavior and attention span of shoppers engaging in online shopping under time pressure were investigated in the present study. The remainder of this paper is structured as follows. Section 2 reviews relevant literature and describes hypotheses about online shopping behavior. Section 3 introduces the method of this study. The findings are presented in Section 4 and their implications and suggestions are discussed in Section 5. Finally, Section 6 concludes this study and outlines this paper’s contributions.

## 2. Background and Hypotheses

Time pressure is an influential factor in consumer behavior [21]; it can have a marked influence on decision-making and restrict information-processing ability [22]. Its effects on people’s decision making intensify in the face of information overload [23]. Moreover, Payne et al. [24] found that people progress through a hierarchy of responses as time pressure intensifies. Specifically, shoppers under moderate time pressure become faster and slightly more selective at information processing, whereas those under heavy pressure tend to skim through information superficially without examining every single detail. However, some studies have suggested that time pressure typically prompts decision makers to make decisions and execute decision-making strategies through simple means [25,26], and that people under time constraints can turn to other strategies to facilitate their information processing [27]. In addition, Levy [28] showed that when people were in a hurry, they hastened their decision making. Pieters and Warlop [18] argued that consumers under time pressure will filter some information, accelerate information acquisition, and adjust their information acquisition strategies.

Generally, time pressure reduces visual attention [29]. Although products with newly designed packaging can attract visual attention [30], such products can be neglected by people who perceive their packaging to be overly novel [31], leading to financial loss and even the removal of some products from the shelf [32].

Clement et al. [6] noted that consumers under time pressure tend to focus on certain products and brands, as well as their characteristics. Studies conducted in brick-and-mortar stores have found that the timing of purchases made under time pressure is similar to those made when not under time pressure. This indicates that consumers do not select certain products due to time pressure; they either make decisions in a matter of seconds when they need to identify familiar information quickly from a pool of information [33], or adjust their search strategies to concentrate on the design features of brands [18]. Accordingly, this study argues that shoppers under time pressure focus on fewer products to facilitate their product search on shopping websites, and that they adjust their search strategies to identify the salient design features of products; this information-processing strategy reflects the stimulation of the brain in a top-down fashion. Based on the aforementioned argument, Hypotheses 1 and 2 were formulated as follows:

**Hypotheses** **(H1).**
*Shoppers view fewer products when shopping under time pressure than they would when shopping not under time pressure.*


**Hypotheses** **(H2).**
*Shoppers focus more on renowned brands when shopping under time pressure than they would when shopping under no time pressure.*


Our ability to focus on the task at hand is a key element in efficient information processing and our attention is easily distracted by novel events or changes in the stimulus environment [34]. Bettman et al. [35] maintained that attention changes occur because of reflexive reactions to threats such as time pressure. In a study by Ordonez and Benson [27], subjects dealing with decisions under time pressure adopted different decision-making strategies to accelerate their information-processing speed. Zur and Breznitz [36] also argued that decision makers typically spend less time viewing information when they are under time pressure, indicating that under such circumstances they may change their decision-making strategies and thereby change their level of attention. Therefore, Hypothesis 3 was formulated as follows:

**Hypotheses** **(H3).**
*Under time pressure, shoppers engaging in online shopping are less attentive than those not under time pressure.*


In the United Kingdom, approximately 70% of consumers who enter grocery stores have incomplete purchase intentions [37]. Previous research [38] has shown that 85% of consumers do not handle commodity items while shopping and 90% of consumers view only the covers of commodity items. Furthermore, consumers tend to purchase products they like after simply viewing them; such actions occur most frequently during online shopping [39]. Brands with sophisticated designs and noticeable visual elements (e.g., product names, logos, layouts, and slogans) can make a deep impression on consumers [40,41,42].

From a cognitive neuropsychological perspective, visual attention can be expressed in terms of orientation-attention and discover-attention. Orientation-attention is a parallel and non-selective pre-attentive search process that enables a considerable amount of information to be processed efficiently and simultaneously. Discover-attention is a serial search process of sequentially searching for information details on the packaging of a product (e.g., textual content and caution labels). In the view of Perkins [43], orientation-attention is the primitive stage of attention, whereas discover-attention enables the complete understanding of a commodity. Neither cognitive system can be distinguished easily in real-world contexts other than shopping [44]. Generally, consumers depend on slow, serial search processes [45]. The presence of branded products and previous online shopping experiences can facilitate their search. Thus, when they have to make purchase decisions in a short time frame or if they intend to purchase renowned products, they tend to simplify their search on shopping websites. Clement et al. [6] assumed that a comprehensive understanding of product catalogs and experiences of shopping at physical stores can expedite product searches, although their findings showed that consumer product searches in brick-and-mortar contexts were facilitated not by their familiarity with product catalogs, but by their understanding of the way products were displayed in-store. However, this study argues that product catalogs on shopping websites differ from those of physical stores; hence, they might facilitate online product searches. Accordingly, Hypotheses 4 and 5 were formulated as follows:

**Hypotheses** **(H4).***Familiarity with product catalogs on shopping websites can reduce product search time during shopping*.

**Hypotheses** **(H5).**
*Experience using other shopping websites can reduce product search time during shopping.*


## 3. Methods

We aimed to understand the effect of time pressure on consumer browsing behavior and attention level with an EEG (electroencephalogram) biosensor cap with regard to branded products on online shopping website. To verify our hypotheses, we conducted a laboratory study on a real-world website. Participants were recruited and assigned to two time-pressure levels (the presence or absence of time-pressure situations). They were assigned a purchasing task and instructed to browse on Taobao (a famous Chinese shopping website). We used an EEG biosensor cap to track the attention level of the participants as they browsed products on the webpage. Upon completion of the experiment, the participants were given a $20 gift card as our token of appreciation for their time and effort. 

### 3.1. Electroencephalogram (EEG) Technique

According to the traditional model of control, physiological systems self-regulate their activity to preserve steadiness by reducing fluctuations around a homeostatic equilibrium point. By contrast with this view, a wide bulk of evidence has recently been provided that several physiological time signals exhibit intrinsic fractal fluctuations. Indeed, heartbeat, respiration, gait rhythm, dynamics of neurotransmitter release, electromyography, and brain activity reveal similar temporal patterns over multiple time scales [46]. In an active postsynaptic neuron, a negative voltage between neural dendrites and other locations along the neuron is generated. Within a small brain compartment in which dendritic structure are parallel and follow a main direction, such a situation can be modelled as a current dipole generating an electromagnetic field. Both electrical potentials and magnetic fields, generated from the dipole in this compartment, can be measured non-invasively by sensors located on or close to the scalp. The technology that measures electrical potentials is called electroencephalography [47].

In this study, a light NeuroSky EEG biosensor cap (i.e., MindWave Mobile as shown in Figure 1) was used for measuring the attention and meditation levels of shoppers. The MindWave Mobile is a portable, wireless hardware cap developed by NeuroSky Company (Taipei, Taiwan). Crowley et al. [48] have evaluated a similar NeuroSky EEG biosensor cap (i.e., Mindset) to measure the attention and meditation levels of a subject in practice. MindWave Mobile outputs 12 bit raw-brainwaves (3–100 Hz) with a sampling rate of 512 Hz and EEG power spectra (alpha, beta, etc.). The detected waves were interpreted by eSense™ (NeuroSky’s proprietary algorithm for characterizing mental states, Windows version v1.2.3) to indicate each subject’s mental state when they were shopping online. For each different mental state (i.e., attention, meditation), the meter value from eSense™ is reported on a relative scale of 1 to 100. On this scale, a value between 40 and 60 at any given moment in time is considered “neutral”. A value from 60 to 80 is considered “slightly elevated”, and may be interpreted as levels being possibly higher than normal. Values from 80 to 100 are considered “elevated”. Similarly, a value between 20 and 40 indicates “reduced”, while a value between 1 and 20 indicates a “strongly lowered” level of each different mental state.

In addition, meditation is considered as a promising technique for body and mind regulation. Meditation plays an important role at physical, mental, and spiritual levels. EEG measures the brain activity useful to recognize the emotional states. EEG has excellent resolution at the millisecond scale, and is superior to positron emission tomography (PET) and functional magnetic resonance imaging (fMRI) [49]. Crowley et al. [48] have evaluated the use of NeuroSky’s Mindset headset to measure the attention and meditation levels of a subject in practice. Thus, analyzing shoppers’ attention and meditation level when shopping under time pressure or not is straightforward.

### 3.2. Subjects and Tasks

To address the research questions, a sample of 30 participants who had online shopping experience was recruited via convenience sampling. While wearing an EEG cap, the subjects performed tasks of product purchases on Taobao. They used the catalogs, listed recommendations, and keywords provided by the website and purchased products of their preference. The experiment was conducted in a quiet laboratory with a laptop computer to ensure that the subjects would not be disturbed. The EEG biosensor cap was connected with Bluetooth to a laptop to record subjects’ brainwaves and obtain experimental data.

### 3.3. Experimental Procedure

An Institutional Review Board (IRB) proof was conducted (Approval No.: NCKU-HREC-E-104-101-2) before this experiment. All subjects were asked to complete a questionnaire regarding their mental and physical states and whether they had ever shopped on any shopping websites, and to sign an informed consent form. 

To reduce any discomfort from wearing the cap, which would affect the experimental results, the subjects were given some time (approximately 3 min) at the beginning of the experiment to accustom themselves to the cap. During the experiment, each subject purchased 10 specific items in odd and even number from the 20 best-selling products sold on Taobao (Figure 2) while under time pressure (10 min) and not under time pressure (unlimited time) conditions. The sample size was comparable with in-store studies of shopper navigation and the number of purchases was also comparable with that used in past in-store work. Table 1 presents the top 20 products on Taobao shopping website at the time of the experiment and their respective item numbers. This experiment was aimed at investigating whether the subjects focused on certain products or renowned brands when under time pressure. The renowned brands were defined according to the American Marketing Association, World Brand Lab and other related websites. The shopper’s choices, durations, visual attention and site navigation were recorded using our own designed program. After the experiment, each subject completed a questionnaire regarding the number of years of experience they had in using shopping websites. Data collected from the questionnaire were analyzed to determine whether familiarity with shopping website product catalogs and experience using shopping websites facilitated the subjects’ purchase behavior in online shopping.

## 4. Results

To test the proposed hypotheses, the experimental data on the subjects’ online shopping behavior were analyzed through the paired t-test to compare the test outcomes between different test conditions and attention types. Descriptive statistics and paired *t*-test results are shown below.

### 4.1. Demographic Data

The demographic data in Table 2 show that the sample comprised 15 men and 15 women who had experience using shopping websites. The subjects were 21–30 years old and all held a bachelor’s degree. Most of the subjects had extensive experience using the Internet (>10 years, 83.4%; 5–10 years, 13.3%; <1 year, 3.3%) and reported that they had used Ruten (86.7%), Yahoo! (80%), PChome (53.3%), Taobao (26.7%), and Amazon (10%) for online shopping. Regarding the frequency with which they used online shopping websites, 46.8% reported that they used them 1–5 times per year, 26.6% reported using them 5–10 times per year, and 26.6% used them >10 times per year. For the number of years of experience using online shopping websites, 50% had 1–5 years of experience, 33.3% had 5–10 years of experience, and 16.7% had >10 years of experience.

### 4.2. T-Test Results

#### 4.2.1. Shoppers Viewed Fewer Products under Time Pressure

The mean of products viewed by all 30 subjects was 28.78 (standard deviation (SD): ±11.986), and the mean time spent on purchasing the 10 assigned product items was 9.097 min (SD: ±3.9072). An independent t-test revealed a non-significant gender difference in the number of products viewed, as shown in Table 3, where SD stands for standard deviation and SE stands for standard error. The F-value was non-significant at 0.958 > 0.05 for the under time-pressure condition and at 0.960 > 0.05 for the not under time-pressure condition. Thus, an equal-variances test was conducted for the under time-pressure condition (*t* = 0.323, *p* = 0.749 > 0.05) and the not under time-pressure condition (*t* = −0.643, *p* = 0.525 > 0.05). No significant difference was observed between the numbers of products viewed by the male and female participants, regardless of the condition. Accordingly, the data were subjected to further analysis.

Table 4 tabulates the descriptive statistics for the number of products viewed under both conditions, and Table 5 presents the paired *t*-test results. The *p* value was 0.001 indicates a significant difference in the number of products viewed between two conditions; specifically, the subjects under time-pressure condition viewed more products.

In addition, regarding the EEG analysis, a paired *t*-test was conducted on the attention and meditation levels of the subjects when shopping under time pressure and not under time pressure conditions; the corresponding descriptive statistics and paired t-test results are presented in Table 6 and Table 7, respectively. The *p*-value for the difference between the attention levels was 0.008 and the difference between the meditation levels was 0.572 (Table 7). These results indicate that under time-pressure and not under time-pressure conditions, the subjects differed significantly in attention but not in meditation. Their attention during online shopping was weaker while under time pressure.

#### 4.2.2. Shoppers Focused more on Renowned Brands while under Time Pressure

This section discusses whether shoppers focus more on renowned brands when under time pressure. Of the 10 assigned product items to be purchased, 1.31 products were from renowned brands on average (SD: ±0.8658). An independent *t*-test revealed a no significant gender difference in the number of branded products purchased (Table 8). The *F*-value was not significant at 0.806 > 0.05 for the under time-pressure condition and at 0.259 > 0.05 for the not shopping under time-pressure condition. Thus, an equal-variances test was conducted for under the time-pressure condition (*t* = 0.638, *p* = 0.529 < 0.05) and not shopping under time-pressure condition (*t* = −0.475, *p* = 0.638 > 0.05). No significant difference was observed between the numbers of branded products purchased by the male and female subjects, regardless of the condition.

A paired *t*-test was conducted to examine whether deducting the number of renowned brand products purchased with the under time-pressure condition from those purchased with the not under time-pressure condition would yield a result greater than zero. Descriptive statistics and the paired *t*-test results are shown in Table 9 and Table 10, respectively. The *p*-value was 0.001 (Table 10), which indicates a significant difference in the number of renowned brand products purchased between the two conditions; specifically, fewer products were purchased under time pressure.

#### 4.2.3. Familiarity with Product Catalogs on Shopping Websites Facilitated Product Searches during Shopping

When shopping with the not under time-pressure condition, the subjects were asked to purchase five products under the condition that the subjects were unfamiliar with the product catalogs, and five others under the condition that the subjects were familiar with the product catalogs. The products are shown in Table 11. The amount of time that the male (ID a, b, c,… with symbol ●) and female (ID A, B, C,…with symbol ▲) subjects spent on product searches is depicted by Figure 3, in which the **x**-axis denotes the amount of time spent searching for products in catalogs that the subjects were familiar with, and the *y*-axis represents the amount of time they spent searching for products in catalogs that they were unfamiliar with.

An independent t-test was conducted to determine the significance level of the gender difference in the product search times (Table 12). The F-value was not significant at 0.057 > 0.05 for catalog familiarity and at 0.075 > 0.05 for catalog unfamiliarity. Thus, an equal-variances test was conducted for the catalog familiarity condition (*t* = −2.550, *p* = 0.017 < 0.05) and the catalog unfamiliarity condition (*t* = −1.745, *p* = 0.092 > 0.05). A significant gender difference was observed in the product search times under the catalog familiarity condition but not under the catalog unfamiliarity condition. Accordingly, the data were subjected to further analysis.

A paired *t*-test was conducted to examine whether deducting the product search time under the catalog unfamiliarity condition from that under the catalog familiarity condition would yield a result greater than 0. Descriptive statistics and the paired *t*-test results for the product search times are presented in Table 13 and Table 14, respectively. The *p*-value was 0.000 (Table 14), which indicates a significant difference in the product search times between the catalog familiarity and unfamiliarity conditions; specifically, longer search times were observed under the catalog unfamiliarity condition as expected.

Moreover, the number of product web pages surfed under the catalog familiarity condition was compared with that under the catalog unfamiliarity condition (Figure 4). An independent *t*-test was used to determine the significance level of the gender difference in the number of product web pages surfed (Table 15). The F-value was non-significant at 0.733 > 0.05 under the catalog familiarity condition and at 0.511 > 0.05 under the catalog unfamiliarity condition. Thus, an equal-variances test was conducted for the catalog familiarity condition (*t* = 0.173, *p* = 0.864 > 0.05) and catalog unfamiliarity condition (*t* = −1.434, *p* = 0.163 > 0.05). No significant gender difference was observed in the number of product web pages surfed between the male and female subjects, regardless of the catalog familiarity or unfamiliarity condition. Accordingly, the data were subjected to further analysis.

The number of product web pages surfed was compared between the catalog familiarity and unfamiliarity conditions. The descriptive statistics and paired *t*-test results are presented in Table 16 and Table 17, respectively. The *p*-value was 0.004, which indicates a significant difference in the number of product web pages surfed between the two conditions; specifically, more pages were surfed under the catalog unfamiliarity condition as expected.

Testing for significant differences in attention and meditation levels between the catalog familiarity and unfamiliarity conditions was undertaken. The descriptive statistics and paired *t*-test results are presented in Table 18 and Table 19, respectively. The *p*-value for attention was 0.007 and the *p*-value for meditation was 0.946 (in Table 19). These results indicate a significant difference between the catalog familiarity and unfamiliarity conditions in the attention levels of the subjects but not in their meditation levels. Notably, the subjects with catalog unfamiliarity demonstrated weaker attention during product searches than those with catalog familiarity.

#### 4.2.4. More Experience Using Shopping Websites Facilitated Product Search

Before the experiment, all the subjects were asked to complete a questionnaire that included items about their number of years of experience using shopping websites. A survey of Taiwanese online shoppers conducted by Foreseeing Innovative New Digiservices (which operates under the Institute for Information Industry, a non-governmental organization promoting the development of Taiwan’s information industry) revealed that most shoppers have more than 5 years of online shopping experience. Accordingly, in this study, those with more than 5 years of experience were defined as “more experienced online shoppers,” and those with less than 5 years of experience were defined as “less experienced online shoppers.” In this survey of our participants, the more experienced online shoppers spent an average of 9.93 min (SD: ±4.2937), compared with their less experienced counterparts, who spent an average of 11.75 min (SD: ±4.8990) in purchases. Therefore, the null hypothesis is “There is no significant difference in the product search times between more experienced and less experienced online shoppers.” Table 20 presents the independent *t*-test results for the product search times between the more experienced and less experienced online shoppers. The test statistic was *t* = 1.073 and *p* = 0.293 > 0.05. Therefore, the difference in product search times between the more experienced and less experienced online shoppers was not significant.

## 5. Discussion and Implications

After experiment and hypothesis testing, the findings of this study and their implications are summarized as follows.

### 5.1. Under Time Pressure, Shoppers Focus on Fewer Products during Online Shopping

This finding corresponds with previous studies that have shown that shoppers under time pressure tend to hasten their product selection [50,51], expedite their information searches to reduce time spent processing information [25,36], or concentrate on specific brands and desired product attributes when making purchase decisions [18]. Therefore, comparing the results of the previous studies revealed that online shopping behavior under time pressure has not changed much despite the rapid development of the Internet and growing use of online shopping services over the past decade.

Moreover, Iyer [22] showed that customers with sufficient time for shopping but without shopping lists are inclined to make more purchases. Dhar and Nowlis [50] noted that online stores typically offer discounts for rush purchases. Ahituv et al. [13] suggested that under time pressure, decision makers tend to make decisions and execute decision-making strategies through simple means. Discounts promote consumption and, under time constraints, they can prompt quick purchase decisions. Most shopping website operators offer limited-edition products that can only be purchased by applying for a membership within certain periods. Such promotional campaigns, which feature a limited number of products, can prompt shoppers to hasten their product searches and make quick purchase decisions. Shopping website operators can launch similar campaigns to encourage consumers shopping under time pressure, such as promotions and special offers, to boost sales and reduce the server load.

### 5.2. Shoppers Engaging in Online Shopping Focus More on Renowned Brand Products When Not under Time Pressure

When shopping under time pressure, specific characteristics of products typically draw the attention of consumers [26,36,52]. New characteristics added to a product often become essential factors that affect consumers’ purchase decisions [53,54]. Distinctiveness exerts stronger effects on product purchase decisions when consumers are under time pressure [50]. Contrary to these previous studies, this study found that subjects did not pay more attention to renowned brand products when they were under time pressure. The result indicated that the renowned brands characteristic of a product are not equivalent to product distinctiveness or novelty, which attract consumer attention under time pressure. In other words, only when a consumer is not under time pressure do they have adequate time to select renowned brand products. This can be attributed to renowned brand products typically being high-priced, prompting consumers to give more consideration prior to making a purchase than they would give on non-renowned brand products. Moreover, Mitchell and Greatorex [55] found that consumers tend to shop at reputable stores to reduce the risk of purchasing low-quality products. Huang et al. [56] showed that brand awareness is crucial for reducing purchase risk. Accordingly, shoppers tend to pay more attention to searching for renowned brand products when not under time pressure to reduce this risk. According to the aforementioned arguments, shopping website operators can sell more renowned brand products during non-promotional periods to encourage consumption and improve customer trust. They can also offer their own-brand products during their time-limited promotions to boost their profits.

### 5.3. Under Time Pressure, Shoppers Engaging in Online Shopping are Less Attentive

The EEG measurement results of this study in Section 4.2.1 reveal that shoppers engaging in online shopping are more attentive when not under time pressure, probably because they adopt the depth strategy under this condition, enabling them to focus on the content and depth of products they view [57]. Thus, shopping website operators are suggested to improve the content and depth of their products during non-promotional periods to enhance consumer trust and approval indirectly. They can also encourage changes in consumer behavior through promotional activities, which can prompt shoppers to process information more quickly [27].

### 5.4. Catalog Familiarity Reduces Product Search Times

Hoque and Lohse [58] found that user-friendly user interfaces can facilitate product searches. Sharpe and Staelin [59] observed that people typically prefer spending less time on familiarizing themselves with the layout of a website. This study showed that consumers familiar with product catalogs of a website spend less time on product searches; this finding accords with those of previous studies in a brick-and-mortar store. Mccarthy and Aronson [60] suggested that browsing functionality on webpages should be designed in a manner that renders the pages easy to navigate. Elliott and Speck [61] remarked that convenient browsing, simple interfaces, and well-organized frameworks of websites can enhance operation ease of use and improve user experience. Other authors have identified that consumers may leave a website quickly if they felt the information was useless [62] or had to complete a hard task online [63]. To avoid this unwanted outcome, shopping website operators should avoid making substantial changes to their user interfaces; such changes can confuse consumers, lengthening their product search times. If the operators intend to refine their websites while reducing the likelihood of deterring customers, they should consider hosting online scavenger hunt-type games so that participants can accustom themselves to the product catalogs on the websites.

### 5.5. Shoppers Unfamiliar with Product Catalogs on Shopping Websites are Less Attentive

Numerous people base their decisions on their past actions [64,65] and inevitably repeat those actions. In psychology, this type of behavior is known as “familiarity” [66]. Soderlund [67] and Payne et al. [68] have argued that familiarity with the shopping environment affects purchase intention; specifically, shoppers with higher familiarity with the environment are more likely to make purchases [69,70] and choose products [71,72]. In familiar shopping environments, shoppers depend on their long-term memory; by contrast, in unfamiliar environments, they rely largely on external messages such as visual stimuli [73]. Moreover, stimuli in unfamiliar environments can attract attention [74], and shoppers at brick-and-mortar stores they are unfamiliar with are highly attentive during shopping [75]. However, the EEG measurement results of this study indicate that shoppers familiar with product catalogs on shopping websites are more attentive during shopping, which is different from Garling et al. [73] and Ashby et al. [76]. Shopping on websites is different from in brick-and-mortar stores. Thus, how consumer attention is affected during online shopping necessitates further research. According to these findings, shopping website operators can retain the original layout of their product catalogs to focus consumer attention ontheir products, which can subsequently facilitate more immediate purchases.

### 5.6. Product Search Times are Not Significantly Shorter for More Experienced Shoppers

Whereas Clement et al. [6] showed that shoppers with more brick-and-mortar shopping experience are more efficient at product searches, this study found no notable reduction in product search times among the more experienced online shoppers. These findings are perhaps due to the differences between the physical and virtual shopping environments. Furthermore, Daly [77] noted that positive attitudes and satisfaction strongly affect online shopping intentions. Swaminathan et al. [78], Deighton [79], and Lepkowska-White et al. [80] have maintained that significant browsing convenience can make information searches easier on websites and improve their popularity. Jin and Park [81] found that less experienced online shoppers have higher perceived risks of shopping websites, whereas their more experienced counterparts tend to focus on the services offered by the websites. They also asserted that service quality becomes increasingly critical for consumers as their transaction relationships with sellers mature. Thus, website operators can improve the security and service quality of their websites (instead of investigating whether their customers have shopped elsewhere online) to strengthen their customer relations and attract new business.

## 6. Conclusions

This study investigated the behavior of consumers shopping online under time pressure, hypothesizing that (1) shoppers under time pressure focus on renowned brand products and are attentive, (2) shoppers with more online shopping experience are more efficient at shopping, and (3) familiarity with product catalogs facilitates product searches. However, the results show that shoppers under time pressure view fewer products and those not under time pressure focus on renowned brand products. In addition, shoppers are less attentive when under time pressure. Moreover, more online shopping experience does not significantly reduce product search times, indicating that the presence of renowned brand products on online shopping websites can lower shoppers’ perceived product risks. Shoppers unfamiliar with product catalogs on shopping websites are less attentive. Furthermore, shoppers under time pressure tend to hasten their purchase decisions and browse fewer product pages. The findings and implications of this study may contribute to relevant academic research and online shopping businesses.

## Figures and Tables

**Figure 1 brainsci-09-00301-f001:**
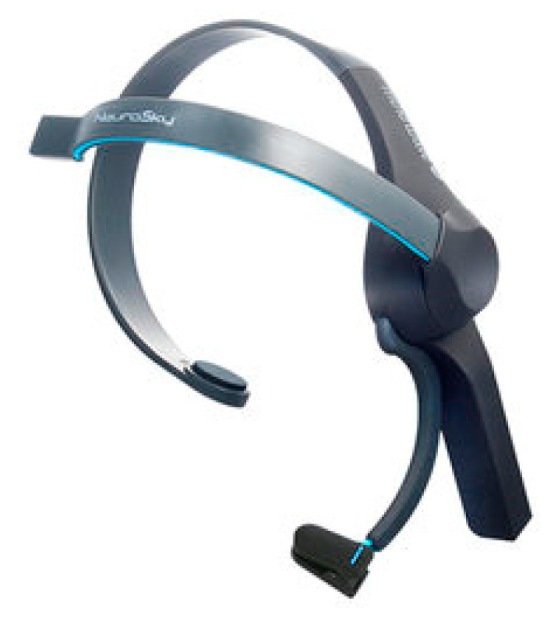
MindWave Mobile, NeuroSky.

**Figure 2 brainsci-09-00301-f002:**
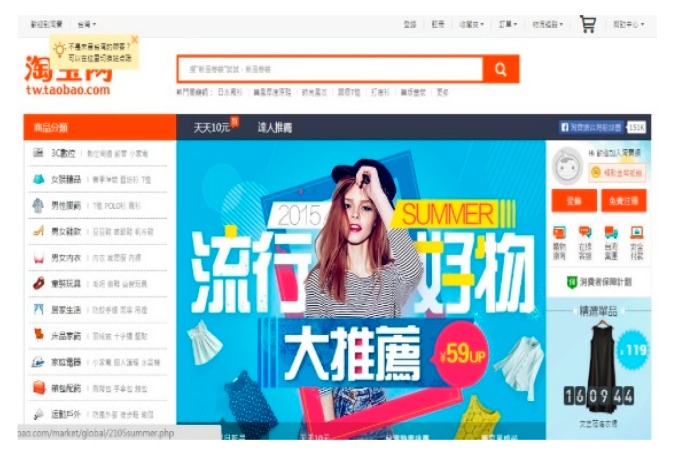
Taobao home page.

**Figure 3 brainsci-09-00301-f003:**
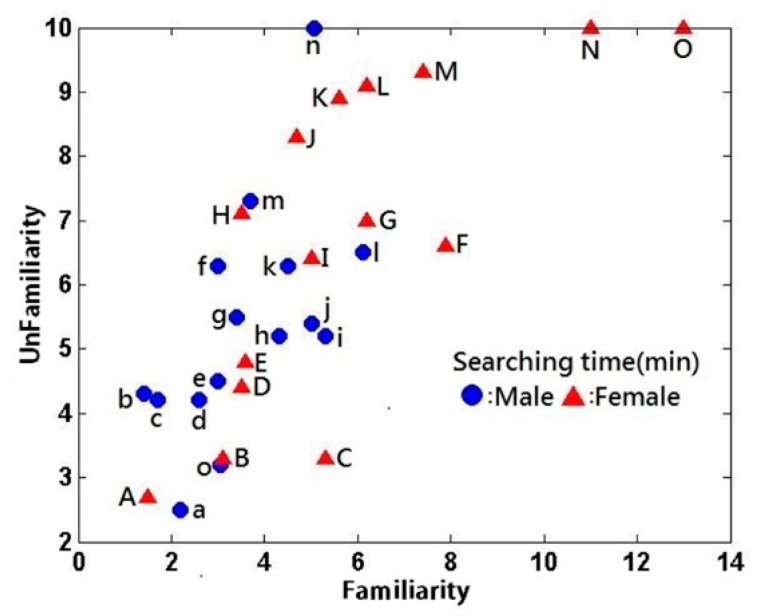
Product search time in minutes.

**Figure 4 brainsci-09-00301-f004:**
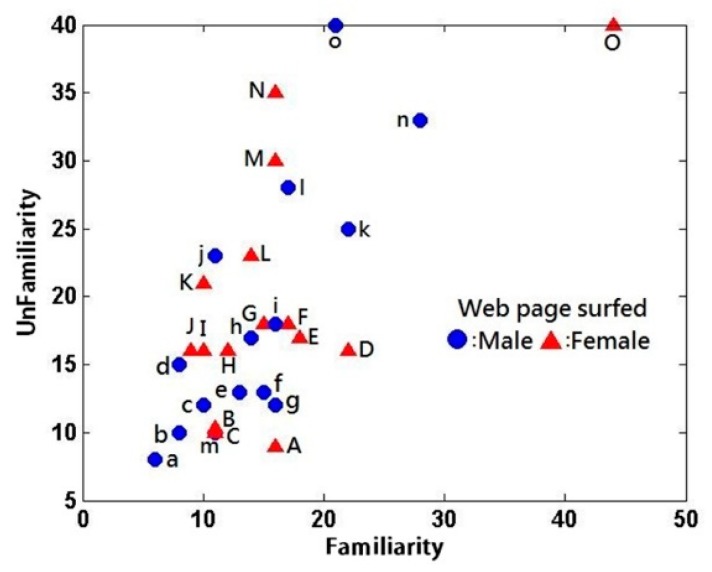
Number of product web pages surfed (familiarity vs. unfamiliarity).

**Table 1 brainsci-09-00301-t001:** Top 20 products on Taobao and assigned items.

Condition	Under Time Pressure	Not Under Time Pressure
Item1	Sweater coat	Dress
Item2	Women’s footwear	Mobile phone
Item3	Shirt	Sweater
Item4	Windbreaker	Children’s apparel
Item5	Wallpaper	iPhone
Item6	Wall sticker	Purse
Item7	Men’s footwear	Cotton-padded clothing
Item8	Thermos	Dr. Martens boots
Item9	Spring and autumn dresses	Spring and autumn women’s bottoming shirt
Item10	Watch	Tablet computer

**Table 2 brainsci-09-00301-t002:** Demographic statistics.

Item	Variable	*N*	%
Sex	M	15	50
	F	15	50
Age	21–31	30	100
Highest level of education	Graduate	30	100
How many years of experience do you have using the Internet?	<1 year	1	3.3
	5–10 years	4	13.3
	>10 years	25	83.4
Have you shopped online before?	Yes	30	100
Which of these online shopping website(s) have you purchased items from before?	Ruten	26	86.7
	PChome	16	53.3
	Yahoo	24	80
	Taobao	8	26.7
	Amazon	3	10
How frequently do you use shopping websites each year?	1–5 times	14	46.8
	5–10 times	8	26.6
	>10 times	8	26.6
How many years of experience do you have using shopping websites?	1–5 times	15	50
	5–10 times	10	33.3
	>10 times	5	16.7

**Table 3 brainsci-09-00301-t003:** Independent *t*-test results for gender differences in the number of products viewed.

Independent Variable	*n*	Mean	SD	SE	*df*	*t*	*p*-Value
Under time pressure (male)	15	23.867	4.580	1.182	28	0.323	0.749
Under time pressure (female)	15	23.333	4.466	1.153			
Not under time pressure (male)	15	32.200	14.178	3.660	28	−0.643	0.525
Not under time pressure (female)	15	35.733	15.867	4.097			

**Table 4 brainsci-09-00301-t004:** Descriptive statistics for the number of products viewed.

Condition	*n*	Mean	Median	Min	Max	SD
Under time pressure	30	23.600	29	14	37	4.453
Not under time pressure	30	33.967	22	18	84	14.893

**Table 5 brainsci-09-00301-t005:** Paired *t*-test results for the number of products viewed between two conditions.

Independent Variable	*n*	Mean	SD	SE	*p*-Value
Under time pressure—Not under time pressure	30	−10.366	15.082	2.753	0.001 ***

*** *p* < 0.001.

**Table 6 brainsci-09-00301-t006:** Descriptive statistics for attention and meditation levels.

Condition	*n*	Mean	Median	Min	Max	SD
Under time pressure (attention)	30	46.941	52.38	33.85	60.48	6.216
Not under time pressure (attention)	30	50.615	53.49	37.93	64.7	6.931
Under time pressure (meditation)	30	56.127	58.26	46.71	64.74	4.453
Not under time pressure (meditation)	30	55.702	56.94	49.2	69.04	14.893

**Table 7 brainsci-09-00301-t007:** Paired *t*-test results for attention and meditation levels.

Independent Variable	*n*	Mean	SD	SE	*p*-Value
Under time pressure—Not under time pressure (attention)	30	−3.673	7.097	1.295	0.008 **
Under time pressure—Not under time pressure (meditation)	30	0.424	4.060	0.741	0.572

** *p* < 0.01.

**Table 8 brainsci-09-00301-t008:** Independent *t*-test results for gender differences in the number of renowned brands.

Independent Variable	*n*	Mean	SD	SE	*df*	*t*	*p*-Value
Under time pressure (male)	15	1.067	0.883	0.228	28	0.638	0.529
Under time pressure (female)	15	0.867	0.833	0.215			
Not under time pressure (male)	15	1.600	0.632	0.163	28	−0.475	0.638
Not under time pressure (female)	15	1.733	0.883	0.228			

**Table 9 brainsci-09-00301-t009:** Descriptive statistics for renowned brand products purchased.

Condition	*n*	Mean	Median	Min	Max	SD
Under time pressure	30	0.967	1	0	3	0.850
Not under time pressure	30	1.667	1	0	3	0.758

**Table 10 brainsci-09-00301-t010:** Paired *t*-test results for the number of products purchased.

Independent Variable	*n*	Mean	SD	SE	*p*-Value
Under time pressure—Not under time pressure	30	−0.7	0.987	0.180	0.001 ***

*** *p* < 0.001.

**Table 11 brainsci-09-00301-t011:** Taobao’s best-selling products purchased not under time pressure.

Condition	Catalog Unfamiliarity	Catalog Familiarity
1	Dress	Purse
2	Mobile phone	Cotton-padded clothing
3	Sweater	Dr. Martens boots
4	Children’s apparel	Spring and autumn women’s bottoming shirt
5	iPhone	Tablet computer

**Table 12 brainsci-09-00301-t012:** Independent *t*-test results for gender differences in the product search times.

Independent Variable	*n*	Mean	SD	SE	*df*	*t*	*p*-Value
Familiarity (male)	15	3.649	1.386	0.357	28	−2.550	0.017
Familiarity (female)	15	5.901	3.127	0.807			
Unfamiliarity (male)	15	5.415	1.796	0.463	28	−1.745	0.092
Unfamiliarity (female)	15	6.847	2.620	0.676			

**Table 13 brainsci-09-00301-t013:** Descriptive statistics for the product search times.

Condition	*n*	Mean	Median	Min	Max	SD
Unfamiliarity	30	6.331	5.50	2.59	10.66	2.295
Familiarity	30	4.998	3.10	1.44	13.53	2.819

**Table 14 brainsci-09-00301-t014:** Paired *t*-test results for the product search times (familiarity vs. unfamiliarity).

Independent Variable	*n*	Mean	SD	SE	*p*-Value
Unfamiliarity—Familiarity	30	1.356	1.789	0.326	0.000 ***

*** *p* < 0.001.

**Table 15 brainsci-09-00301-t015:** Independent *t*-test results for gender in the number of web page surfed.

Independent Variable	*n*	Mean	SD	SE	*df*	*t*	*p*-Value
Familiarity (male)	15	15.467	8.854	2.286	28	0.173	0.864
Familiarity (female)	15	15.000	5.567	1.437			
Unfamiliarity (male)	15	16.733	8.655	2.234	28	−1.434	0.163
Unfamiliarity (female)	15	21.400	9.155	2.364			

**Table 16 brainsci-09-00301-t016:** Descriptive statistics for the number of product web page surfed.

Condition	*n*	Mean	Median	Min	Max	SD
Unfamiliarity	30	19.067	9.50	8	40	9.070
Familiarity	30	15.233	13.50	6	44	7.271

**Table 17 brainsci-09-00301-t017:** Paired *t*-test results for the number of web page surfed.

Independent Variable	*n*	Mean	SD	SE	*p*-Value
Unfamiliarity—Familiarity	30	–3.833	6.664	1.216	0.004 **

** *p* < 0.01.

**Table 18 brainsci-09-00301-t018:** Descriptive statistics for attention and meditation.

Condition	*n*	Mean	Median	Min	Max	SD
Familiarity (attention)	30	52.914	55.44	41.71	71.63	8.741
Unfamiliarity (attention)	30	48.797	51.30	32.94	58.9	6.464
Familiarity (meditation)	30	56.169	55.44	48.01	72.28	5.143
Unfamiliarity (meditation)	30	56.241	58.82	49.16	66.44	4.885

**Table 19 brainsci-09-00301-t019:** Paired *t*-test results for attention and meditation levels.

Independent Variable	*n*	Mean	SD	SE	*p*-Value
Unfamiliarity—Familiarity (attention)	30	4.117	7.820	1.427	0.007 **
Unfamiliarity—Familiarity (meditation)	30	−0.072	5.791	1.057	0.946

** *p* < 0.01.

**Table 20 brainsci-09-00301-t020:** Independent *t*-test results for the product search times.

Independent Variable	*n*	Mean	SD	SE	*df*	*t*	*p*-Value
Less experienced shoppers	16	11.754	4.899	1.224	28	1.073	0.293
More experienced shoppers	14	9.937	4.293	1.147			

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
