# Peer review of "Exploring Shopper’s Browsing Behavior and Attention Level with an EEG Biosensor Cap"

_brainsci, 2019, doi:10.3390/brainsci9110301_

Round 1

Reviewer 1 Report

Point 1: In summary, Dong-Her Shih and colleagues conducted a study about online shoppers under two conditions: under pressure and not under pressure. The results indicated a reduction of attention under time pressure. Moreover, shoppers unfamiliar with product catalogs on shopping websites are less attentive. Authors suggest that marketers may need to find strategies to increase shopper’s attention. 

Although the study is well formulated
and results and conclusions are clear and well presented, I think that some improvements to the introduction and methods sections can improve the overall paper quality:

Specifically, it is not clear to me why, in a real case context, a consumer should browse on line products under pressure.  Please clarify and add a brief clarification in the introduction.

Response 1:

Thank you for the reviewer’s valuable comment. As suggested, the authors have rewritten the introduction section and add a paragraph to clarify why consumer should browse on line products under pressure as below to improve the overall paper quality in this new revised version.

Generally, retailers want consumers to spend more time shopping, browsing, and searching for products in the hope that some purchases happen. On the other hand, consumers may want to restrict their duration of stay on websites due to perceived time/convenience loss risk [8]. Previous research on discussing purchasing decision process has assumed the time pressure on shoppers. For example, Lin & Wu [9], found that under the conflicts of product attributes or choice set alternatives, no time pressure and high time pressure will increase the proportion of consumer no-attitude on judgment task and no-choice option on choice task. Vermeir and Kenhove [10] suggested that high time-pressured consumers search less for coupons and products with a promotion. Rieskamp and Hoffrage [11] demonstrate that compare to low time pressure, under high time pressure, individual accelerate the search for information, using less information, and staying focused on the most important features. Liu et al. [12] indicate that when shopping online under time pressure, participants’ observation length and count for browsing products with high brand awareness were respectively longer and higher than those for browsing products with low brand awareness. However, when they shopped online without time pressure, no difference between products with high and low brand awareness levels was observed.

The authors have also rewritten the methods section and add a paragraph as below to improve the overall paper quality in this new revised version. Thank you for the reviewer’s effort.

  1. Methods

We aimed to understand the effect of time pressure on consumer browsing behavior and attention level with an EEG biosensor cap with regard to branded products on online shopping website. To verify our hypotheses, we conducted a laboratory study on real world website. Participants were recruited and assigned to two time pressure levels (the presence or absence of time pressure situations). They were assigned a purchasing task and were instructed to browse on Taobao (a famous Chinese shopping website). We used an EEG biosensor cap to track the attention level of the participants as they browsed products on the webpage. Upon completion of the experiment, the participants were given a $20 gift card as our token of appreciation for their time and effort.

Point 2: Lines 116-122 seems to me “out of place”. I would suggest to remove these lines and add at the beginning of the methods section a brief summary about EEG technique.  See for example (as reference) the introduction section of [1] and/or [2].

 [1] Croce, P.; Zappasodi, F.; Marzetti, L.; Merla, A.; Pizzela, V. & Chiarelli, A. M.

Deep Convolutional Neural Networks for feature-less automatic classification of Independent Components in multi-channel electrophysiological brain recordings

IEEE Transactions on Biomedical Engineering, Institute of Electrical and Electronics Engineers (IEEE), 2018, 1-1

 [2] Croce, P.; Quercia, A.; Costa, S. & Zappasodi, F.

CIRCADIAN RHYTHMS IN FRACTAL FEATURES OF EEG SIGNALS

Frontiers in Physiology, Frontiers, 2018, 9, 1567

Response 2:

Thank you for the reviewer’s valuable comment. As suggested, Lines 116-122 have been moved to the bottom of Section 3.1 under a brief summary about EEG technique and these two suggested citations have been added at the top of section 3.1 in this new revised version. Thank you for the reviewer’s effort.

Point 3: In section 3.2 what is an IRB proof? Please specify.

Response 3:

Thank you for the reviewer’s valuable comment. “IRB stands for Institutional Review Board. IRBs review and monitor how a research study will be conducted to ensure the study does not cause unreasonable risks to participants.” As suggested, the authors have added this paragraph in Acknowledgement section to clarify the confusion in this new revised version. Thank you for the reviewer’s effort.

Point 4: Moreover, it would be appropriate to describe how the attention and meditation scales are computed. Provide the mathematical details of such scale.

Response 4:

Thank you for the reviewer’s valuable comment. Unfortunately, the authors cannot obtained the mathematical details of the attention and meditation scales due to the proprietary reason. The reference is listed below. Thank you for the reviewer’s effort.

eSense™ is a NeuroSky’s proprietary algorithm for characterizing these mental states which have powerful capabilities when integrated into education, sports coaching, meditation, and other mind-controlled games. To calculate eSense™, the NeuroSky ThinkGear technology amplifies the raw brainwave signal, removes the ambient noise and muscle movement. Then the eSense algorithm is applied to the remaining signal, resulting in the interpreted eSense meter values. Later, the values can be translated into effect or exported for further analysis.

[Ref] Gudinavičius, Arūnas. (2016). Towards understanding the differences between reading on paper and screen: measuring attention changes in brain activity. Libellarium: journal for the research of writing, books, and cultural heritage institutions. 9. 10.15291/libellarium.v9i1.240.

Point 5: Line 198, which software?

Response 5:

Thank you for the reviewer’s valuable comment. The authors are sorry for the writing typo, the sentence should be “The shopper’s choices, durations, visual attention and site navigation were recorded using our own designed program“. The authors have corrected this mistake in this new revised version. Thank you for the reviewer’s effort.

Minor points:

Point 6: Lines 46-47: not clear, rephrase.

Response 6:

Thank you for the reviewer’s valuable comment. As suggested, the authors have deleted Lines 46-47 and written second paragraph in Introduction section to make it more clearly in this new revised version. Thank you for the reviewer’s effort.

Reviewer 2 Report

Dear authors,

Thank you for your manuscript. The manuscript makes for an interesting read but requires significant work on several key areas. Most importantly, it is unclear why consumers are set out to shop under pressure? The rationale is unclear and does not illuminate the purpose of the study. Secondly, the methods section needs to be detailed (i.e. software and version used, acronyms explained). References need to be added to justify the methods used. Last but not least, extensive editing of English language needs to be done for grammar.

Response:

Thank you for the reviewer’s valuable comment. As suggested, “why consumers are

set out to shop under pressure?” has been described and rewritten at second

paragraph in Introduction section. The authors have also rewritten the methods

section and add a paragraph as below to improve the overall paper quality. In

addition, software and version used, acronyms explained are also included at section 3.1 in this new revised version. Thank you for the reviewer’s effort.

Methods

We aimed to understand the effect of time pressure on consumer browsing

behavior and attention level with an EEG biosensor cap with regard to branded

products on online shopping website. To verify our hypotheses, we conducted a

laboratory study on real world website. Participants were recruited and assigned to

two time pressure levels (the presence or absence of time pressure situations). They

were assigned a purchasing task and were instructed to browse on Taobao (a famous

Chinese shopping website). We used an EEG biosensor cap to track the attention level

of the participants as they browsed products on the webpage. Upon completion of the

experiment, the participants were given a $20 gift card as our token of appreciation

for their time and effort.

The authors had paid a great attention to the editing of English. Therefore, the

authors had cost almost 1,000 US dollars (New Taiwan dollar $29,591) to found a

professional English editing company (Wallace academic editing), as attachments

below, to edit English writing before submitting this manuscript. The first version of

this manuscript is “online shopping behavior of consumers when under time

pressure”, and the authors have changed the title to “Explore shopper’s browsing

behavior and attention level with an EEG biosensor cap“ due to more clearly in its

content. However, the authors have forward the reviewer’s comment to this

professional English editing company (Wallace academic editing) and ask them for

help. Wallace academic editing only promise editing the modified part the authors

did in this new revised version but not all of them. Therefore, if the reviewer is

unsatisfied with this new revised version, the authors promise to find another

professional English editing company to make it better in English reading. Thank

you for the reviewer’s effort.

The authors are very appreciated for all the reviewer’s efforts.

Round 2

Reviewer 1 Report

Please have the paper edited by a native English speaker or a professional editor.

Author Response

English editing has been done and editing proof is as attachment. Thank you for the reviewer's comment and effort.

Reviewer 2 Report

All my concerns have been addressed. 

Author Response

English editing has been done and editing certificate is as attachment. Thank you for the reviewer's comment and effort.
